



# Resolving the size of ice-nucleating particles with a balloon deployable aerosol sampler: the SHARK

Grace C. E. Porter[1,2], Sebastien N. F. Sikora[1], Michael P. Adams[1], Ulrike Proske[1,3], Alexander D. Harrison[1], Mark D. Tarn[1,2], Ian M. Brooks[1] & Benjamin J. Murray[1]

[1]School of Earth and Environment, University of Leeds, Leeds LS2 9JT, UK
[2]School of Physics and Astronomy, University of Leeds, Leeds LS2 9JT, UK
[3]Institute for Atmospheric and Environmental Sciences, Goethe University Frankfurt, Frankfurt am Main, Germany

*Correspondence to*: Grace C. E. Porter (ed11gcep@gmail.com) and Benjamin J. Murray (b.j.murray@leeds.ac.uk)

**Abstract.** Ice-nucleating particles (INPs) affect cloud development, lifetime and radiative properties, hence it is important to know the abundance of INPs throughout the atmosphere. A critical factor in determining the lifetime and transport of INPs is their size, however very little size-resolved atmospheric INP concentration information exists. This is especially so in the free troposphere. Here we present the development and application of a radio-controlled payload capable of collecting size-resolved aerosol from a tethered balloon for the primary purpose of offline INP analysis. This payload, known as the SHARK (Selective Height Aerosol Research Kit), consists of two complementary cascade impactors for aerosol size-segregation from 0.25 to 10 µm, with an after-filter and top stage to collect particles below and above this range at flow rates up to 100 L min⁻¹. The SHARK also contains an optical particle counter to quantify aerosol size distribution between 0.38 and 10 µm, and a radiosonde for the measurement of temperature, pressure, GPS altitude, and relative humidity. This is all housed within a weatherproof box, can be run from batteries for up to 11 h and has a total weight of 9 kg. The radio control and live data link with the radiosonde allow the user to start and stop sampling depending on meteorological conditions and height, which can, for example, allow the user to avoid sampling in very humid or cloudy air, even when the SHARK is out of sight. While the collected aerosol could, in principle, be studied with an array of analytical techniques, this study demonstrates that the collected aerosol can be analysed with an off-line droplet freezing instrument to determine size-resolved INP concentrations, activated fractions and active site densities, producing similar results to those obtained using a standard PM₁₀ aerosol sampler when summed over the appropriate size range. Test data is presented from four contrasting locations having very different size resolved INP spectra: Hyytiälä (Southern Finland), Leeds (Northern England), Longyearbyen (Svalbard), and Cardington (Southern England).

## 1 Introduction

Atmospheric ice-nucleating particles (INPs) are not well understood, with knowledge of their concentration, sources, temporal variability, transport and size in its infancy (Kanji et al., 2017; Murray et al., 2012). This is of importance because clouds between 0 °C and around −35 °C can exist in a supercooled liquid, mixed-phase (ice and water) or glaciated (ice only) state depending in part on the presence or absence of INPs (Kanitz et al., 2011; Vergara-Temprado et al., 2018). In the absence of INP, cloud droplets can supercool to below ∼-35°C (Herbert et al., 2015), but INP can trigger freezing at much higher temperatures (Kanji et al., 2017). These particles usually





have concentrations that are orders of magnitude smaller than cloud condensation nuclei (CCN), and have a
disproportionate impact on clouds because the nucleated ice crystals grow rapidly and precipitate out (Lohmann,
2017; Murray, 2017). In a shallow cloud, heterogeneous ice nucleation can result in dramatic reductions in cloud
albedo by removal of supercooled liquid water (Storelvmo, 2017; Vergara-Temprado et al., 2018), whereas in
deep convective clouds it can influence a web of microphysical processes in a complex way (Deng et al., 2018;
Kanji et al., 2017; Rosenfeld et al., 2011). Hence, a greater understanding of INP lifetime, transport and
distribution in the vertical profile is needed in order to better understand and model cloud processes and their
response to a changing climate.

The size of an aerosol particle significantly affects its lifetime and therefore transport in the atmosphere, with
particles of a few hundred nanometres having a lifetime of weeks in the free troposphere, whereas particles of 10
$\mu$m have a lifetime of only hours (Jaenicke, 2007). It has been generally thought that the larger an aerosol particle,
the more likely it is to serve as an INP (Pruppacher and Klett, 1997), but the lifetime of coarse mode aerosol
particles decreases rapidly with increasing size. Consistent with larger particles being better ice nucleators,
parameterisations of INPs in the atmosphere have been proposed wherein the INP concentration is related to the
concentration of aerosol particles larger than 0.5 $\mu$m (Demott et al., 2015; DeMott et al., 2010; Tobo et al., 2013).
However, most atmospheric measurements of INPs report the sum of INPs below some threshold size set by an
inlet or size cut, specified by the aerosol sampler used. For instance, DeMott et al. (2017) provides a comparison
between a selection of instruments for the collection and subsequent INP analysis of aerosol, where the aerosol
samplers have either a defined size cut-off or have collection efficiencies that decrease in magnitude above a
defined size. Nevertheless, there are examples of field studies in which INPs have been size-resolved (Berezinski
et al., 1988; Creamean et al., 2018b; Huffman et al., 2013; Mason et al., 2016; Reicher et al., 2018; Santachiara
et al., 2010; Si et al., 2018; Welti et al., 2009). These studies generally show that while the fine mode aerosol
particles are more abundant, coarse mode aerosol particles often contribute more to the INP population. In
addition, the activated fraction ($n_n$) of coarse mode aerosol is usually greater than fine mode aerosol. However, in
some field studies (Mason et al., 2016; Si et al., 2018), fine aerosol sometimes contributes more to the INP
population than the coarse mode. Therefore, there is a need to determine INP sizes when quantifying atmospheric
INP concentrations, as size is important for transport and lifetime and is therefore required to accurately model
global INP populations.

Measurements of INPs in and above the boundary layer are crucial to understanding the contribution of local
sources to the ice-nucleating activity in clouds, compared to transported aerosol. Aircraft measurements (e.g. Price
et al., 2018; Rogers et al., 2001) and mountaintop observatories (e.g. Conen et al., 2015) have been used to quantify
INP populations above the boundary layer. For example, it has been shown that there are differences in the INP
concentrations measured when in and out of the boundary layer at the High Altitude Research Station Jungfraujoch
(Switzerland) (Conen et al., 2015; Lacher et al., 2018). While these measurements are undoubtedly useful,
mountaintop measurements are only possible in locations with sufficiently tall yet accessible mountains, and
aircraft sampling is expensive and not necessarily possible in remote regions. It is therefore essential that
instrumentation is available that can be used to sample aerosol at selected altitudes (including ground level) in
order to determine INP concentrations throughout the vertical profile. Unmanned aerial vehicles (UAVs) are



becoming more widely used in atmospheric science; these allow the collection of aerosol at altitude at significantly
lower cost than with manned aircraft, but are limited by relatively short battery lives of 10s of minutes and
potential propeller interference (Jacob et al., 2018; Villa et al., 2016).

Tethered kite and balloon systems have historically been used to make atmospheric measurements and collect
aerosol samples with much longer sampling times (many hours are readily achievable) at altitudes up to 2 km and
5 km for tethered balloons and kites respectively (Armstrong et al., 1981; Balsley et al., 1998). An advantage of
a balloon or kite system is that an instrument can be held at a chosen altitude for many hours without the balloon
interfering with measurements, as the instrument can be suspended on a line many meters below the balloon. They
can also stay inflated and in use for periods of many weeks, making them ideal for longer campaigns in remote
environments. A new instrument called the Honing On VERtical Cloud and Aerosol properTies (HOVERCAT)
(Creamean et al., 2018a) provides the capability to sample aerosol for subsequent INP analysis on a tethered
balloon or UAV, allowing both variable altitudes and static collection of non-size resolved aerosol smaller than
10 μm at 1.2 L min$^{-1}$. In the past, aerosols have been size-segregated using cascade impactors on a tethered balloon
system (Hara et al., 2013; Reagan et al., 1984), but balloon-borne cascade impactor systems have not yet been
adapted for the purpose of size-resolved INP analysis. The downsides of balloon-based platforms include the need
for wind speeds below around 64.4 km h$^{-1}$ to avoid damage to the balloon, and the possibility of 'icing' of the
balloon and lines when deployed in a cold and humid environment, which could add to the weight of the payload
and cause the system to sink, or fall slowly. Nevertheless, balloon and kite-borne measurements remain a valuable
way to obtain continuous, high resolution measurements over a period of many hours in a single location at a
range of altitudes.

In this paper, the design, testing and operation of a payload named the Selective Height Aerosol Research Kit
(SHARK) is presented. It consists of two separate cascade impactor systems, operating at 9 and 100 L min$^{-1}$, for
the size-sorting of ambient aerosol particles from 0.25 to 10 μm, with an after-filter and top stage to collect
particles below and above this range for offline INP (or other) analysis. The SHARK also features an optical
particle counter (OPC) and a radiosonde, which provides real-time measurements of relative humidity (RH),
temperature, Global Positioning System (GPS) altitude and pressure. Weighing 9 kg, the payload is suitable for
use with a 21 m$^3$ or larger tethered balloon such as in Figure 1 a where the SHARK is shown in-flight. The use of
a tethered balloon and a high-capacity battery allow aerosol to be collected for up to 11 h at a user-selected altitude.
**2      The design and development of the SHARK**
**2.1      Instrument description**
The SHARK, shown in Figure 1, comprises two cascade impactors and corresponding pumps, alongside an OPC
(OPC-N2, Alphasense, UK) and radiosonde (S1H2-R, Windsond, Sweden), all mounted within a weatherproof
enclosure with a tail fin to orient it into the wind. A photograph of the internal components of the SHARK are
shown in Figure 1b. The two cascade impactors were employed to collect particles across different size bins:
Impactor 1 from 0.25-2.5 μm and Impactor 2 from 1-10 μm. Impactor 1 is a cascade impactor (U.S. Patent No.
6,786,105, Sioutas, SKC, UK), which requires a flow rate of 9 L min$^{-1}$ and operates with a portable pump (Leland



Legacy, SKC, UK). Impactor 2 is also a cascade impactor (MSP Model 128, TSI, USA), which requires a flow
rate of 100 L min$^{-1}$ at a pressure drop of 0.6 kPa (Marple et al., 1991; Misra et al., 2002), and for which a radial
flow impeller (Radial Blower U51, Micronel, UK) was used in reverse as a lightweight pump (~120 g). In order
to provide RH, temperature, GPS altitude and pressure data in real-time, the sensors and transmitter from a
radiosonde were integrated into the system. The OPC measured aerosol size distributions, which were saved in
the on-board memory. Servo-controlled caps covered the sample inlets and outlets to reduce contamination during
ascent and descent, as well as to protect the components from cloud water. The operation of the SHARK
components was controlled remotely via a radio link using an Arduino microcontroller board; once the SHARK
was at the desired altitude according to the constantly transmitting radiosonde, the inlet caps opened 10 s prior to
the pumps and OPC starting in order to initiate aerosol sampling and monitoring. The payload components,
including the servo inlet covers and Arduino control boards, were powered by a 5000 mAh battery (4S 14.8 V
LiPo, Overlander, UK). The components were assembled into the SHARK payload with the static (i.e. no wind)
weight budget of 10 kg for a 21 m$^3$ balloon (Skyhook Helikite, Allsopp Helikites Ltd., UK) in mind, hence the
SHARK weighs 9 kg when fully instrumented.

The cascade impactors allow for the collection of size-segregated aerosol (further details are provided in Section
2.2) onto thin films (0.25 mm thickness) for subsequent off-line analysis, which can be used alongside information
about the aerosol size distributions obtained via the OPC and atmospheric conditions from the radiosonde. Our
initial focus concerns the analysis of the ice-nucleating properties of the collected aerosol, but an array of
analytical techniques could be applied to characterise the size-selected aerosol, including mass spectrometry,
DNA analysis, scanning electron microscopy (SEM) and transmission electron microscopy (TEM) (Ault and
Axson, 2017; Garcia et al., 2012; Huffman et al., 2013; Laskin et al., 2018).
**2.2    Size-segregated collection of aerosol**
Two separate cascade impactors were installed, each operating over different size ranges. This enabled size-
resolved aerosol sampling onto substrates across both the fine and coarse modes at high flow rates, while keeping
power consumption low enough to be run from batteries. Single impactor systems designed to operate across the
accumulation and coarse modes simultaneously require a relatively large pressure drop that would typically
require a prohibitively large (and heavy) pump and battery for this application.

Impactor 1 sorts aerosol into five size categories: <0.25 μm (this size bin is defined by the impactor after-filter
and is hereafter referred to as 1$a$), 0.25-0.5 μm (from stage 1$b$), 0.5-1.0 μm (from stage 1$c$), 1.0-2.5 μm (from
stage 1$d$), and >2.5 μm (from stage 1$e$). The size categories $b$ to $e$ correspond to the impactor stages where the 50
% collection cut-off diameter (d50) is the lower bound of each bin. The size bins and collection efficiencies for
each impactor were digitised from data provided by the manufacturers, (Misra et al., 2002; Product Information
Sheet - MSP) and are shown in Figure 2. Several collection substrates were tested by Misra et al. (2002), and the
dataset from the Teflon substrates was chosen to represent Impactor 1 here as that substrate most closely resembled
those used in this study. For Impactor 1, the particles were collected on 25 mm diameter filters of pore size 0.05
μm (Nuclepore Track-Etched Membrane polycarbonate filters, Whatman, UK). Filters were used as impactor
substrates rather than films since they have very low background contamination and are easier to obtain. Size


category 1*a* corresponds to an after-filter situated after Impactor 1, which comprised a 47 mm diameter
polycarbonate filter with a pore size of 5 µm (Nuclepore Track-Etched Membrane) to maintain the flow rate. The
collection efficiency of the after-filter was estimated to be 50-100 % at 0.25 µm and below (Soo et al., 2016).
Impactor 2 collected aerosol particles into three size categories: 1.0-2.5 µm (2*d*), 2.5-10 µm (2*e*), and >10 µm
(2*f*), also illustrated in Figure 2. 75 mm diameter filters of pore size 0.05 µm (Nuclepore Track-Etched Membrane
polycarbonate filters) were used in Impactor 2. An after-filter could not be used with this impactor since its
inclusion increased the required pressure drop to beyond what the pump could supply at 100 L min$^{-1}$.

A further benefit of using these two impactors in tandem is that, in the size ranges where they overlap of 1.0-2.5
µm (stage *d*) and 2.5-10 µm (stage *e*), the impaction efficiencies are very similar, allowing a direct comparison
between the two impactors in this size range. The stages are labelled *a* through *f* for the smallest to largest impactor
stage sizes (including the after-filter), such that 1*d* and 2*d* refer to stage *d* (1.0-2.5 µm) on Impactors 1 and 2,
respectively (see Figure 2). Background runs were produced by placing the substrates in the SHARK as if setting
up to sample, before removing and analysing them as normal to determine the contamination introduced through
the installation and recovery of the substrates.

Particle bounce, the bouncing of particles off the impaction substrate and the collection of these particles on the
lower stages, has previously been identified as a factor that can cause biases when aerosol is collected by cascade
impactors (Cheng and Yeh, 1979; Dzubay et al., 1976). The collection efficiency curves shown in Figure 2 for
Impactor 1 already account for some degree of particle bounce, having been determined experimentally by Misra
et al., (2002) using monodispersed polymer particles on a variety of substrates. However, the efficiency curves
for Impactor 2 are based on theoretical predictions (Rader and Marple, 1985) and so do not account for any bounce
effects. Since two of the stages of Impactors 1 and 2 overlap (stages *d* and *e*), it is possible to comment on the
possible effects, or lack thereof, of particle bounce, based on the results obtained using each of the comparable
stages. This is briefly addressed in section 3.4 where we show good agreement between these two impactors.
**2.3    Size distribution measurements**
The OPC produced binned particle size distributions from 0.38-17 µm every 1.2 s. The OPC was remotely
operated through the use of its serial link via an Arduino microcontroller board. Particle size, surface area and
mass concentration data were produced from the raw OPC data, and these then used to calculate the fraction of
the aerosol that act as an INP (activated fraction, $n_n$), and to weight the INP data to particle surface area or mass,
generating the ice-active site density per surface area ($n_s$) or mass ($n_m$) of aerosol. The particle density used was
1.65 g cm$^{-3}$, as assumed by the OPC software, and they were assumed to be spherical. No correction was made
for the hygroscopic growth of aerosol particles as this required assumptions about the chemical nature of the
particles, and hygroscopic growth effects were minimised by avoiding sampling when the RH was above about
80 % (see next section).
**2.4    Radiosonde data**
Utilising the radio control built into the payload, real-time data informed decisions of when to turn the pumps on
and off to sample. Continuous monitoring of the radiosonde data allows the user to avoid sampling under




conditions where RH approached 100 %, at which point aerosol particles become excessively swollen with water
or activated to cloud droplets. Hence, the influence of hygroscopic growth or cloud droplets on the collected
aerosol could be minimised. The temperature and pressure measurements allowed the volume of air sampled by
the impactors and OPC to be corrected to standard conditions (1 atm at 0 °C).
**2.5    Housing and instrument orientation**
The weatherproof housing consisted of an acrylonitrile butadiene styrene (ABS) polymer box with dimensions of
560 mm x 380 mm x 180 mm (IP67, Fibox). Holes to mount the impactors and OPCs were drilled so that Impactor
2 sat vertically upright and Impactor 1 was oriented 180° to Impactor 2 so that it faced downwards, ensuring that
both impactors were always oriented 90° to the wind. The OPC was at 90° to both impactors and facing towards
the front of the box, into the wind (see Figure 3a-c) See section 2.6 for the rationale of the positioning of the OPC
and impactor inlets. The tail fin, which is mounted to the lid of the box, was designed to keep the SHARK
orientated into the wind, and was fabricated from rigid polyvinyl chloride (PVC) sheet.  Impactor 1 had its own
mounting screws by which it was attached to the box, whilst for Impactor 2 a custom mount was built. Securing
ropes were threaded through reinforced holes in the box and connected via a carabiner for quick and easy
attachment to the balloon instrument line, as seen in Figure 1a. Modular foam was used to keep all components
in place during flight.
**2.6    Inlet sampling efficiencies via particle loss modelling**
Calculation of the particle losses associated with the instrument inlets due to excessive wind speeds in various
configurations were used to inform the design of the SHARK and to minimise sampling biases in higher wind
conditions. The calculations were done using an open source particle loss calculator program in Igor Pro, the
details and assumptions for which are presented Von Der Weiden et al., (2009). The particle loss characteristics
of the impactor and OPC inlets at their required flow rates were calculated for a wind speed of 0 and 24 km h$^{-1}$,
the latter used as a maximum representative wind speed for operation. The wind speeds required for optimum
performance are <8 km h$^{-1}$ for the impactors and OPC, but the system may experience higher wind speeds. Hence,
we use this modelling to guide our choice of positioning of the instrument relative to wind direction in order to
minimise sampling biases at the inlets. The modelling also allows us to better understand which impactor stages
(and OPC size bins) will be most affected by such biases. We make no attempt to correct the measurements for
sampling biases, since this correction itself would carry substantial uncertainty, but used the calculations to inform
us of the best configuration for the various inlets.

The inlet sampling efficiencies in the orientations chosen for the final design of the SHARK are shown in Figure
3. It is important to note that, due to their dissimilar inlet dimensions and operational flow rates, Impactors 1 and
2 are affected differently by the wind. The particle losses for the largest stages of each impactor are the most
affected.  Stages *a* to *d* on both impactors are only minimally affected by losses.  The losses are more significant
in stage e on both impactors, but the losses on 1e are greater than on 2*e* with a 50% cut off at around 5.5 μm and
a negligible sampling efficiency above about 8 μm on 1*e*. These calculations also demonstrate that the losses are
wind-speed dependent, but that in situations where there is significant wind, the results from Impactor 2 will be
less influenced by losses than Impactor 1 at sizes above 2.5 μm






The OPC suffers up to 1.6 times oversampling for 10 µm particles when sampling into 24 km h⁻¹ wind, but when
oriented at 90° to the wind the collection efficiency of >6 µm particles approaches 0 % (see Figure 3c). Therefore,
the OPC has been positioned in the SHARK to be oriented into the wind to ensure data is collected for the whole
size range, with the caveat of a sub-isokinetic oversampling of larger particles.
**3    Results and Discussion**
The SHARK has been deployed at ground level and on a tethered balloon during development and testing at four
locations for the collection and monitoring of aerosol: Cardington (UK), Hyytiälä (Finland), Leeds (UK), and
Longyearbyen (Svalbard). In this section, we present the results for this set of four SHARK deployments to
illustrate the capabilities of the SHARK for quantifying ice-nucleating particle spectra as well as demonstrating
that the technique is consistent with more established methods.
**3.1    Meteorological and aerosol size distribution data from a SHARK flight**
An example of the radiosonde and OPC data that was collected during a SHARK flight is shown in Figure 4. The
data was from a sampling event in the High Arctic in the summer of 2018, during which the meteorological data
from the radiosonde and aerosol particle data from the OPC were collected alongside impactor films for INP
analysis (the INP results will be published elsewhere). Throughout the 4.5 h flight the altitude, humidity and
temperature were closely monitored to inform decisions on sampling. The sampling start and end times are
indicated as solid lines in Figure 4. The SHARK reached 450 m above Mean Sea Level (MSL) and in the last hour
of flight lowered to 350 m due to ice formation on the balloon, instrument and tether. The RH during the flight
was monitored to ensure the SHARK did not sample in humidity approaching saturation; the impactor and OPC
manufacturers' specified thresholds for the components is 95 % RH, but we aim to only sample with the RH below
this value (~80 %) in order to reduce the influence of hygroscopic growth on aerosol size. After sampling was
stopped, the SHARK was brought down to ground level, resulting in the humidity rising. The ability to stop the
sampling during the flight meant the impactors were covered and the pumps turned off during the descent and so
did not sample the more humid environment. The ambient temperature was monitored alongside the dewpoint
temperature to follow the surface inversions. The temperature inversion was used to determine where to stabilise
the SHARK and begin sampling, as sampling was desired above the surface inversion for this run.

The total particle counts per 1.38 s interval from the OPC are shown in Figure 4d. Processing of the OPC data
yielded the results shown in Figure 5 for the particle number ($dN/dlogD_p$), particle surface area ($dS/dlogD_p$) and
particle mass ($dM/dlogD_p$) size distribution data for the sampling period. We present this data to demonstrate that
the OPC produces reasonable data when used facing into wind while suspended from a balloon at altitude.
Unfortunately, there is no direct comparison with other aerosol size distribution measurements at the sampling
location. While the particle number concentration increases roughly linearly with size, the surface and mass
concentration curves have a mode at around 4 µm in Figure 5b and 5c. This is consistent with previous studies
conducted within the boundary layer in the Arctic (Freud et al., 2017; Hegg et al., 1996; Seinfeld and Pandis,

267 2016).




### 3.2    Deriving size-resolved INP concentrations from the SHARK samples


The ability to measure INP concentrations and properties using samples collected via the SHARK was tested by
performing immersion mode droplet freezing assays on the sampled aerosols. Following a flight, impactor films
were removed from both cascade impactors of the SHARK, then each immersed in 5 mL of water and mixed on
a vortex mixer for 5 min to wash the collected particles into suspension (O'Sullivan et al., 2018). This suspension
was then analysed via a droplet freezing assay using the microlitre Nucleation by Immersed Particle Instrument
(μL-NIPI) (Whale et al., 2015), in which 40-50 droplets of 1 μL volume were pipetted onto a hydrophobic glass
slide atop a cold plate. A Perspex shield was placed over the cold stage and $N_2$ gas introduced to purge the chamber
of moisture as the cold plate was cooled to −40 °C at 1 °C min$^{-1}$. The temperatures at which droplets froze were
recorded using video analysis until the entire population had frozen. This allowed the fraction of droplets frozen
as a function of temperature, $f_{ice}(T)$, to be calculated (O'Sullivan et al., 2018; Whale et al., 2015) using the equation
$f_{ice}(T) = N_f / N_t$ , where $N_f$ is the number of frozen droplets at temperature $T$, and $N_t$ is the total number of droplets.
The INP concentration per volume of sampled air as a function of temperature, $[INP]_T$, was then calculated for
each film using $f_{ice}(T)$, according to Equation 1 adapted from (Vali, 1971) to include weighting to the volume of
air sampled:
$$[INP]_T = \frac{-ln(1-f_{ice})}{V_{droplet}} \cdot \frac{V_{wash}}{V_{air}} , \qquad (1)$$
where $V_{droplet}$ is the droplet volume (i.e. 1 μL), $V_{wash}$ is the amount of water into which the filter is immersed to
produce the suspension for analysis (i.e. 5 mL), and $V_{air}$ is the volume of air sampled.

### 3.3    Testing the SHARK INP concentrations against a standard aerosol sampler


In order to test whether the SHARK impactors were sampling in a representative manner, the SHARK was run
concurrently with a filter-based particle sampler (BGI PQ100, Mesa Labs) and which is used as an EPA Federal
Reference Method for $PM_{10}$ (designation no. RFPS-1298-124). This sampler was equipped with a PM10 head and
an optional cyclone impactor which provided a size cut at 2.5 μm. Aerosol was collected onto 0.4 μm pore size
Nuclepore Track-Etched Membrane polycarbonate filters at a flow rate of 16.7 L min$^{-1}$ (i.e. 1 m$^3$ h$^{-1}$). This type
of filter collects particles across the full range of available aerosol sizes, even at sizes smaller than the pore
diameter, with high collection efficiencies (Lindsley, 2016; Soo et al., 2016). These polycarbonate filters have
also been successfully employed in other ice nucleation field measurements (DeMott et al., 2016; Harrison et al.,
2018; Huffman et al., 2013; McCluskey et al., 2016; Reicher et al., 2019; Tarn et al., 2018). These substrates are
known to have a low ice-nucleating ability and allow the collected particles to be released into suspension for
subsequent INP analysis (O'Sullivan et al., 2018). The filters were analysed using the μL-NIPI in the same manner
as for the impactor films collected using the SHARK. The PQ100 filter sampler was deployed alongside the
SHARK in Cardington (UK) and in Hyytiälä (Finland).

In order to compare the SHARK-derived, size-resolved INP data with the results of the $PM_{10}$ or $PM_{2.5}$ PQ100
filter sampler, the INP concentrations determined across the appropriate SHARK size categories were summed.



In Figure 6a, data is presented from Cardington, where the sum of 2*d* and 2*e* from SHARK is compared with the
filter sampler fitted with a $PM_{10}$ head (Impactor 1 was not available during this test). The SHARK was suspended
from a tethered balloon roughly 20 m from the ground, whereas the filter sampler was on the ground (inlet ~150
cm above the surface), where both samplers were within the well-mixed boundary layer. The agreement is very
good apart from two highest temperature points from the filter sampler, but note that the Poisson uncertainties on
these points are substantial and also that the two samplers were separated vertically by 20 m.

We then show data from Hyytiälä in Figure 6b where we compare the INP spectrum from the filter sampler, with
a $PM_{2.5}$ cut-off installed, with the sum of stages 1*b*, 1*c* and 1*d* (the after-filter, stage 1*a* was not used on Impactor
1 in this case). Here, both samplers were positioned within a few metres above the ground. Again, the agreement
between the SHARK and the filter sampler was very good. For both Cardington and Hyytiälä, the smallest
particles (<0.25 μm) were not sampled using the SHARK, but the agreement between the filter sampler and the
SHARK implies that, in these cases, the smallest particles made a minor contribution to the overall INP
population, which is what we would generally anticipate from the literature (Berezinski et al., 1988; Huffman et
al., 2013; Mason et al., 2016; Santachiara et al., 2010; Si et al., 2018; Welti et al., 2009). The consistency between
the SHARK and the filter sampler indicates that there are no major losses of aerosol in the SHARK sampler, at
least relative to the PQ100 filter sampler.

### 323    3.4    Consistency of INP concentrations between SHARK impactors

An example of data from the size-resolved collection and analysis of INPs is shown in Figure 7, from a sampling
run performed in Leeds (UK). The $f_{ice}(T)$ curves for each impactor stage are illustrated in Figure 7a. As discussed
in section 2.2, there are two stages, *d* and *e*, which have similar size cuts on both stages. Using stage *e* as an
example, it can be seen that while the fraction frozen curves for the two samplers are shifted by about 3 °C (Figure
7a), normalising to the volume of air sampled to yield $[INP]_T$ in Figure 7b shows that the INP spectra derived
from stages 1*e* and 2*e* are consistent with one another. Stage 2*e* covers a lower range of INP concentrations than
stage 1*e* by about 1 order of magnitude, because the flow rate through this impactor was more than a factor of
11.1 (100 L min$^{-1}$ / 9 L min$^{-1}$) higher and the probability of collecting rarer INP was increased by this factor. The
agreement between the two impactors indicates that aerosol was collected with no significant losses/enhancements
due to factors like particle bounce or wind observed. Based on the inlet particle loss calculations in Figure 3,
higher losses may have been expected in impactor stage 1*e*, but these are not apparent here.

### 336    3.5    Size-resolved ice-nucleating particle (srINP) spectra at four locations

The derived size-resolved INP (srINP) concentrations for all four test sites are shown in Figure 8 and Figure 9.
Figure 8 shows the INP concentration spectra in the classic form, wherein INP concentrations are plotted against
temperature for each size bin, whereas Figure 9 shows the same data in novel srINP plots to allow more intuitive
comparison of the INP concentration contribution from each stage with respect to temperature. In Figure 9, where
there were measurements from two impactors for the same stage (e.g. *d* and *e*), the INP concentrations were
merged by taking an average at temperature intervals of 0.5°C (also for Figure 6). The colour gradient in Figure



9 represents the temperature dependant concentration for each size bin and the overall steepness of the $d[INP]_T/dT$
curve. The steepness of the INP spectra can be useful in discriminating between different INP species. On
inspection of Figure 8 and Figure 9, it can be seen that the spectra in the four locations have very different
characteristics. Not only does the general shape of the spectra vary, but the size-dependence is also very different
in the four locations. We now discuss the size-resolved INP concentration spectra from these tests, bearing in
mind that these four tests were one-offs and should not be regarded as characteristic of those sampling sites, but
rather illustrative of the importance of making size-resolved measurements.

The first site testing of a prototype of the SHARK in which all of the components were installed was conducted
in Cardington (UK) on the 15[th] of May 2018, but only Impactor 2 was used (see Figure 6a and Figure 8a). The
Cardington site is an airfield, with large areas of grassy land near a main road, and the sampling was conducted
during spring. The SHARK was hung from a tethered balloon roughly 20 m above the ground. The INP spectra
(Figure 8a and 9a) in this location are steep, increasing two orders of magnitude within 2.5 °C, and are centred
around −18 to −20°C; the $[INP]_T$ for 2*f* and 2*e* increases by an order of magnitude in just ~1 °C. The INPs in this
location were dominated by particles greater than 2.5 µm, whereas particles between 1-2.5 µm made a smaller
contribution and show a shallower $d[INP]_T/dT$, seen in Figure 9a as a larger spread of data. We speculate that the
course mode INPs at this site were of biological origin, possibly pollen, based on the size of the INP and the
steepness and temperature range of the spectra being similar to those recorded in laboratory studies of pollen (O
′ Sullivan et al., 2015; Pummer et al., 2012; Tarn et al., 2018).

In Hyytiälä (Finland), a field site in the boreal forest, the INP spectra contrast quite strongly with those in
Cardington (see Figure 6b and Figure 8b). Sampling took place on the 11[th] of March 2018, when the Hyytiälä site
was snow-covered and sampling was performed at the surface (inlet ~150 cm above surface). In this case only
Impactor 1 was used without the after-filter installed. The complex nature of the size-dependence of INP is clear
here. Intriguingly, in this location, the INP concentration was greatest for the smallest stage used (1*b*; 0.25-0.5
µm), and accounted for the majority of the INPs between −17 and −22 °C. The fewest INP came from the next
smallest stage 1*c* (0.5-1 µm), while at temperatures below −23 °C, stage 1*e* contained the majority of the INPs.
These results indicate that the INP spectra are complex, and that concentrations of INPs do not always increase
with increasing size as might be expected. Huffman et al. (2013) reported INP concentration measurements in a
forest ecosystem, where the particles between 1.8 and 5.6 µm enhanced during rain. Hence, as in the present study,
Huffman et al. (2013) showed that INP activity does not always increase with size. The highest INP concentrations
in Hyytiälä were measured for aerosol sizes of 0.25-0.5 µm, and we note that these accumulation mode INPs
would have lifetimes of many days to weeks in the atmosphere and could therefore be transported to locations and
altitudes where they may influence clouds. Clearly, this would be an interesting location for more measurements
with the full SHARK payload to gain further information on the long term INP concentration variations and the
aerosol sizes responsible for them.

The testing in Leeds (UK) used both impactors at ground level with the SHARK suspended from a frame to allow
orientation into wind. The Leeds sampling was conducted within the University of Leeds campus on a patch of
grass on the 7[th] of June 2018 in close proximity to the School of Earth and Environment. In this test the full suite



of impactors and after filters were deployed. It can be seen in Figure 8c that generally, the larger bins contained
more active INP. The only exception to this occurred with the after-filter (< 0.25 μm), which had slightly higher
INP concentrations below about −25 °C than the next two size bins (0.25 – 1.0 μm). As with the measurements in
Hyytiälä, clearly more measurements illuminating the contribution of the smaller particles in similar environments
would be beneficial since the atmospheric lifetime of these fine particles is relatively long. We note that a
substantial proportion of INPs quantified just outside of Leeds in a previous study were heat-sensitive and
therefore most likely of biological origin (O'Sullivan et al. 2018). In the future, conducting heat tests, as well as
using Mass Spectrometry, SEM and DNA analysis with the size-resolved INP samples may help to identify the
INP types in the various size fractions and highlight any differences between size ranges.

The final test was in Longyearbyen (Svalbard) from the 7[th] deck of the icebreaker Oden, 25 m above the surface,
when moored ~200 m from the shore, overnight from the 23[rd] to the 24[th] of September 2018. The full SHARK
payload was used in this case, with both impactors and the after-filter on Impactor 1. The INP spectra in this
location, shown in Figure 8d was quite distinct from the other three locations in that all size fractions contributed
similarly to the INP population and there is a very shallow slope of $\mathrm{dln[INP]}_T/\mathrm{d}T$ (Figure 9d). We detected INPs
at temperatures of up to −10°C with concentrations of around 0.01 INP L$^{-1}$. These high-temperature INP
concentrations are consistent with the summertime measurements reported at other Arctic locations, including
Ny-Ålesund (Svalbard) (Wex et al., 2019). The INP in this region potentially originate from a range of sources.
Tobo et al. (2019) recently reported that dust and biological material from glacial valleys in Svalbard may be an
important source of INPs in the region. We also note that we sampled while the Oden was moored in the port of
Longyearbyen where local pollution sources may have been significant (Zhao et al., 2019).

**3.6    Ice-active surface site density, $n_s(T)$ and the activated fraction, $n_n$**
The addition of size distribution information to the INP concentration spectra allowed the calculation of the
number of active sites per unit surface area, $n_s(T)$ and the activated fraction, $n_n(T)$ of the size resolved samples.
These quantities are determined by weighting the srINP concentrations to the total surface area and the aerosol
number in each size bin, respectively, as shown in Equations 2 and 3.
$$n_s(T) = -\frac{ln(1-f_{ice}(T))}{A_s},$$ (2)
where $A_s$ is the total surface area of the particles per droplet in a μL-NIPI droplet freezing assay. This was
calculated for each impactor size range, using data from the relevant size bins of the OPC data.
$$n_n(T) = -\frac{ln(1-f_{ice}(T))}{N},$$ (3)
where $N$ is the total number of particles sampled during the sampling period in each size category measured by
the OPC.

Calculating the $n_s(T)$ and $n_n(T)$ values from the INP data was only possible for some of the size ranges due to the
sampling ranges of the instrumentation employed. The smallest particle diameter measured by the OPC is 0.38
μm, i.e. above the lower limit of impactor stage 1b, while the largest impactor stage, 2f ( >10 μm) has no defined
upper bound. Therefore, the three bins (i.e. impactor stages) that were used to produce $n_s(T)$ and $n_n(T)$ were c (0.5-



1.0 μm), $d$ (1.0-2.5 μm) and $e$ (2.5-10 μm). The $n_s(T)$ and $n_n(T)$ data were calculated for the field tests in Leeds
and Longyearbyen; data from Cardington and Hyytiälä is not provided as the OPC was not in use at these sites.

The plots of activated fraction shown in Figure 10 are addressed first. For the Leeds sample, there is a difference
in the $n_n(T)$ values between bins $c$ to $e$ (Figure 10a), where the smallest bin is 1-3 orders of magnitude lower than
the largest bin, with the middle bin in the centre of the two. In Longyearbyen (Figure 10b), the $n_n(T)$ for bin $e$ is
about a factor of 10 larger than bin $c$, but bins $c$ and $d$ produce very similar values of $n_n(T)$. Overall, these $n_n(T)$
plots show that the coarse mode aerosol generally have a higher fraction of aerosol that serve as INPs than the
fine mode, but there is variability in the dependence on size between the two samples. In contrast to the $n_n(T)$
values, the size resolved $n_s(T)$ data for both Leeds and Longyearbyen show that the data from the three size
categories are all within a factor of 2-10 (close to our uncertainty estimates). Given the activity of aerosol across
these bins scales with surface area, this data might indicate the same INP species is active across each bin at these
sites.
**4    Conclusions**
This paper describes a lightweight and portable payload, the SHARK, that is capable of collecting size-resolved
aerosol particles alongside measurements of ambient temperature, relative humidity, pressure, GPS coordinates,
aerosol number distribution and aerosol size distribution. The 9 kg payload was designed for use on a tethered
balloon for measurements at user-selected altitudes for up to 11 h via radio controlled instrumentation, but can be
used wherever it can be suspended. During a SHARK flight, the atmospheric conditions the SHARK experiences
can be monitored in real-time via a radiosonde and sampling is controlled remotely, allowing the SHARK to be
held at a user-defined height and to only sample under specific conditions (for instance above the surface boundary
layer).

The SHARK samples aerosol onto filter/film substrates using two cascade impactors to allow aerosol size-
segregation from 0.25 to 10 μm, with an after-filter and top stage to collect particles below and above this range.
One impactor samples at 9 L min$^{-1}$, while the other samples at 100 L min$^{-1}$. The filters were collected here for the
offline analysis of INP concentrations and properties, but they could equally be used for other analyses such as
mass spectrometry, DNA analysis, SEM, TEM and ion chromatography. A comparison of ambient INP
concentrations measured using the SHARK to those measured using PM$_{10}$ and PM$_{2.5}$ aerosol samplers at ground
level demonstrated excellent agreement between the instruments. Field testing was conducted in four locations to
demonstrate the capabilities of the SHARK.

The size resolved INP concentration spectra reveal complex behaviour. For example in Hyytiälä the 0.25-0.5 μm
aerosol size fraction had the most active INP, whereas in Leeds the INP concentration generally decreased with
decreasing particle size. Ambient aerosol size distribution measured using the on-board OPC allowed the
calculation of the activated fraction ($n_n$) and ice-active surface site density ($n_s$) data for the sampled INPs in the
tests at Leeds and Longyearbyen. It was shown that $n_s(T)$ was consistent between 0.5 and 10 μm in these two





locations at the times of sampling. It will be interesting to make similar measurements in other locations in the
future.

Generally, it is expected that larger aerosol are more likely to nucleate ice (Pruppacher, H.R. and Klett, 1997) and
our results are consistent with other size resolved INP measurements which indicate that the size distribution of
INP varies spatially and temporarily e.g. (Mason et al., 2016; Si et al., 2018). Quantifying the size of INP, possibly
in conjunction with other analytical techniques, is a useful means of identifying different INP types and their
sources (Huffman et al., 2013). In addition, knowledge of their size will allow the improved representation of INP
in global aerosol models where size is key determinant of lifetime and transport (Atkinson et al., 2013; Perlwitz
et al., 2015; Vergara-Temprado et al., 2017). Clearly, more systematic and widespread measurements of INP size
is needed in the future in a range of target locations.

The high sample flow rate, choice of low contamination aerosol collection substrates and long sampling durations
mean that the payload is well suited for INP measurements, including those in low aerosol environments and
locations with relatively low INP concentrations (down to below ~0.01 INP L$^{-1}$ and at temperatures down to about
-25°C and below). The SHARK is an accessible tool for quantifying size-resolved atmospheric INP concentrations
through the vertical profile, both within and above the atmospheric boundary layer. This will allow improved
determination of INP sizes, properties, and sources, towards ultimately improving model representations of
atmospheric INP distributions.
**Data availability**
The data sets for this paper will be made publicly available in the University of Leeds Data Repository upon
publication.
**Author contribution**
GCEP led the development of the SHARK, performed the bulk of the experiments and led the writing of the paper.
The initial instrument concept was conceived by GCEP, SNFS and BJM with advice from IMB. The building and
testing of the SHARK and its electrical components was done by SNFS with the assistance of GCEP. The
collection and analysis of field samples was performed by GCEP, MPA, UP, ADH, MDT and IMB. All authors
contributed to the writing of this paper. BJM oversaw this project as part of his MarineIce ERC fellowship.
**Competing interest**
The authors declare that they have no conflict of interest.
**Acknowledgements**
The personnel of Hyytiälä forestry station, the HyIce project team, the Cardington meteorological research unit,
and those aboard the Oden icebreaker during 2018 are sincerely thanked for support during field testing. The





authors thank the European Research Council for funding (H2020 ERC; 648661 MarineIce) and the Natural
Environment Research Council (NERC, NE/M010473/1, NE/R009686/1 ). We are grateful to the EU's H2020
ACTRIS-2 for a mobility grant to access the Hyytiälä forestry station as part of the HyIce project (SMR7 RP3
HyICE18, 654109). Anthony Windross and Stephen Burgess are thanked for help with the fabrication of the
SHARK housing.



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



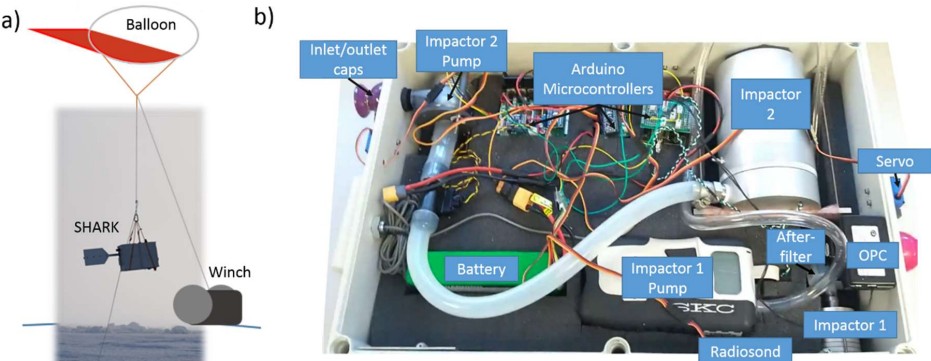

**Figure 1.** The SHARK. (a) The SHARK payload on a tethered balloon connected to ground by a winch. The photograph was taken during deployment in the High Arctic. (b) The components inside the SHARK payload labelled on a photograph. The payload featured a large impactor inlet at the top of the platform for Impactor 2, with the OPC inlet facing the front, and a small impactor inlet at the bottom for Impactor 1. The radiosond was at the bottom of the box, and the outlet valve for the pump system is shown at the back of the SHARK, where the 100 L min$^{-1}$ pump for Impactor 2 vents.





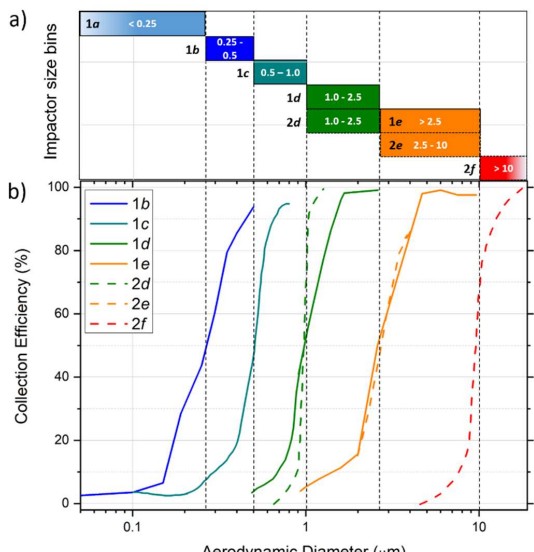

**Figure 2.** Collection efficiencies of each size bin of the two cascade impactors in the SHARK. (a) The size bins for each stage of Impactor 1 and 2 at flow rates of 9 and 100 μL min[-1], respectively. (b) Impactor efficiency curves for each stage. Impactor 1 has four stages (1*b-e*) and one after-filter (1*a*), while Impactor 2 has three stages (2*d-f*). Stages 1*d* and 2*d* as well as 1*e* and 2*e* should be approximately equivalent in terms of the aerosol size ranges collected.

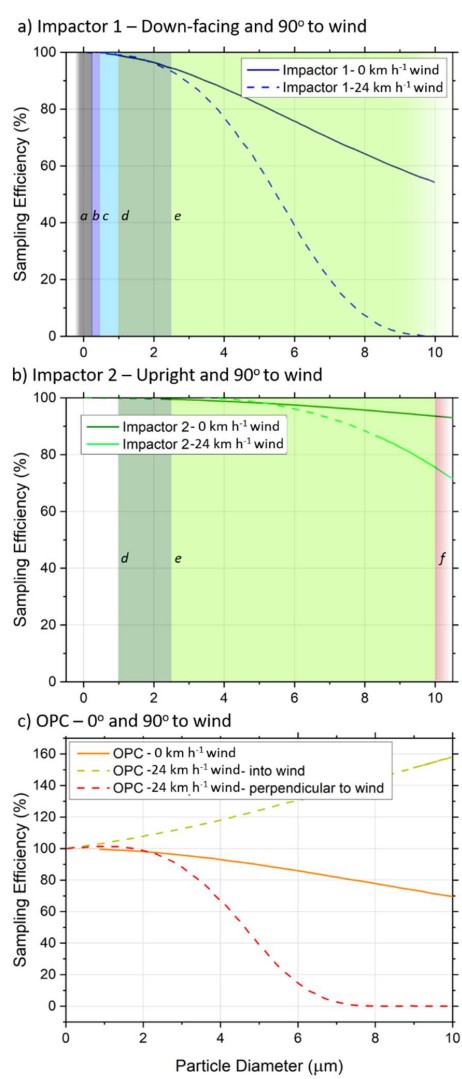

**Figure 3.** SHARK sampling efficiencies (a) The sampling efficiencies of Impactor 1, with and without wind, when sampling at 90° to the wind direction. (b) The sampling efficiencies of Impactor 2, with and without wind, when sampling at 90° to the wind direction. (c) The sampling efficiency of the OPC, with and without wind, when sampling at 0° and 90° to the wind direction (the OPC was deployed at 0° to the wind, based on this calculation). Solid lines denote model predictions within the formulas' validity range, and dotted lines represent approximations (Von Der Weiden et al., 2009).



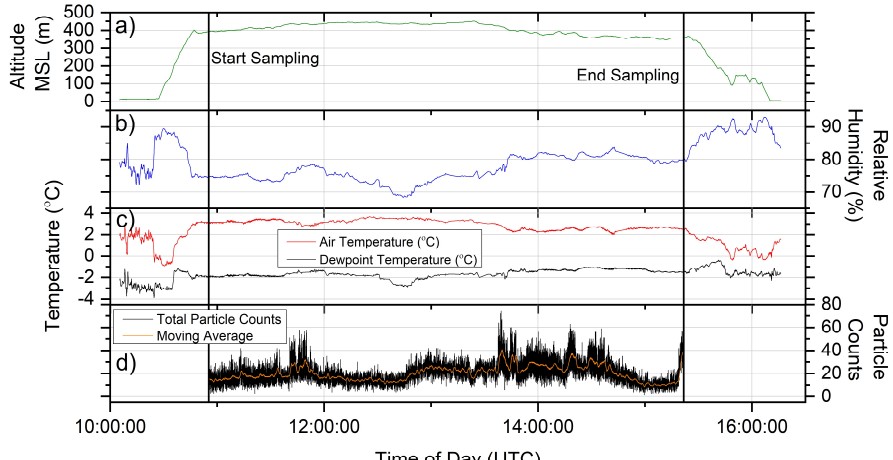

**Figure 4.** Windsond and optical particle counter (OPC) data for a flight during a campaign to the High Arctic. (a) The altitude of the SHARK payload throughout the 4.5 hour flight. The sampling start and end times are indicated as solid lines. The SHARK reached 450 m above Mean Sea Level (MSL) and in the last hour of flight was lowered to 350 m due to ice formation on the balloon, instrument and tether. (b) The humidity during the flight was monitored to ensure the SHARK was not sampling during unfavourable conditions. The SHARK was brought back down to ground level once the sampling had been stopped. (c) The ambient temperature was monitored alongside the dewpoint temperature. (d) Total particle counts throughout the sampling period, as monitored by the OPC.



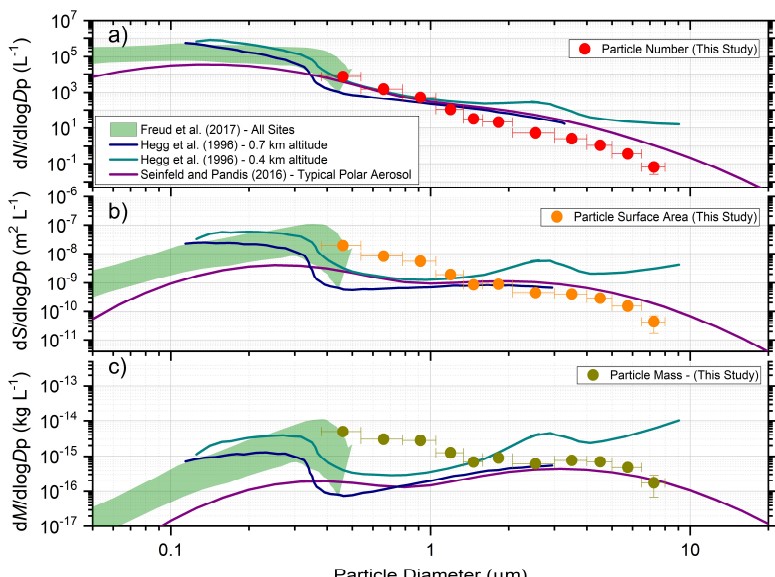

**Figure 5.** OPC a) number, b) surface area and c) mass size distribution data above the surface temperature inversion during a test run of the SHARK suite whilst deployed on a tethered balloon in the High Arctic. Comparisons to previous studies at Arctic sites are shown (Freud et al., 2017; Hegg et al., 1996; Seinfeld and Pandis, 2016). The August aerosol number size distributions for all listed sites in Freud et al., including Zeppelin, Nord, Alert, Barrow and Tiksi are shown. The data from Hegg et al., at two altitudes, 0.7 and 0.4 km are presented. The size distributions from Seinfeld and Pandis are calculated given the parameters for multimode distributions given in Table 8.3.





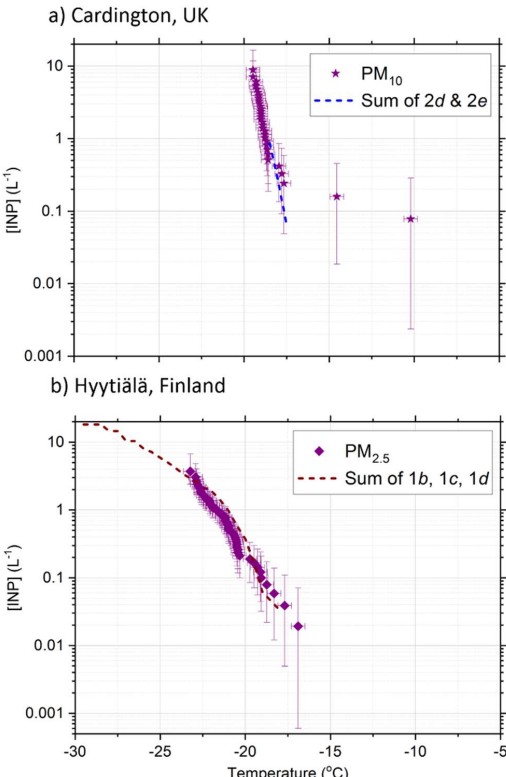

**Figure 6.** The sum of INP concentrations for labelled stages measured at: (a) Cardington (UK) and (b) Hyytiälä (Finland) alongside data from a standard sampler. Cardington data was taken from Impactor 2 whilst on a tethered balloon at 20 m above ground level, and is shown against a $PM_{10}$ sampler at ground level. Hyytiälä data was collected using Impactor 1 at ground level, alongside a $PM_{2.5}$ sampler. The dotted lines indicate the sum of the INP concentrations for the SHARK impactor stages, calculated by weighting $f_{ice}(T)$ to the volume of sampled air, and summing the concentrations in each temperature bin.





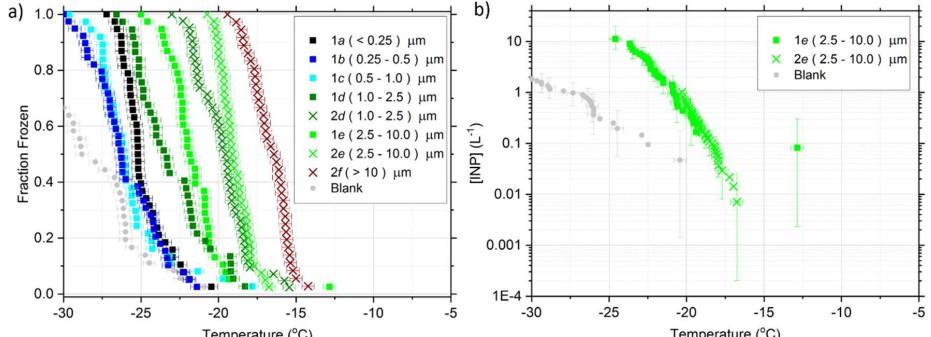

**Figure 7.** Ice-nucleating particle (INP) analysis of samples collected in Leeds (UK) using the SHARK. (a) The fraction of droplets frozen as a function of temperature, $f_{ice}(T)$, for each stage of Impactors 1 and 2. The handling blank is shown in grey. (b) The INP concentrations for stage '$e$' of both impactors (2.5-10 μm), highlighting their excellent agreement.



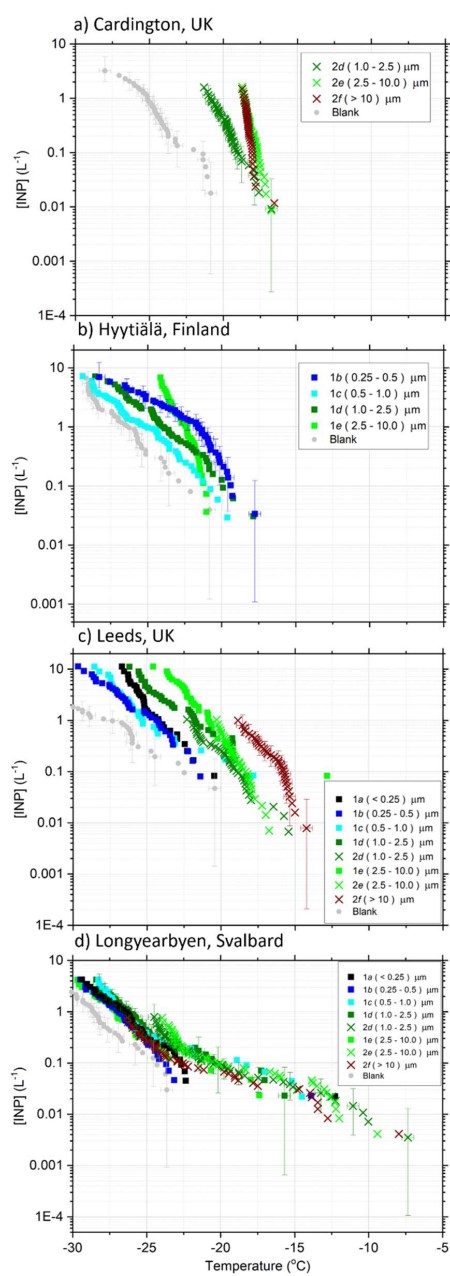

**Figure 8.** INP concentrations determined from each impactor stage of the SHARK at the four testing sites: (a) Cardington (UK), (b) Hyytiälä (Finland), (c) Leeds (UK) and (d) Longyearbyen (Svalbard). Handling blank data, which determine the baseline of the results, are shown in grey. Samples of the error bars are shown.





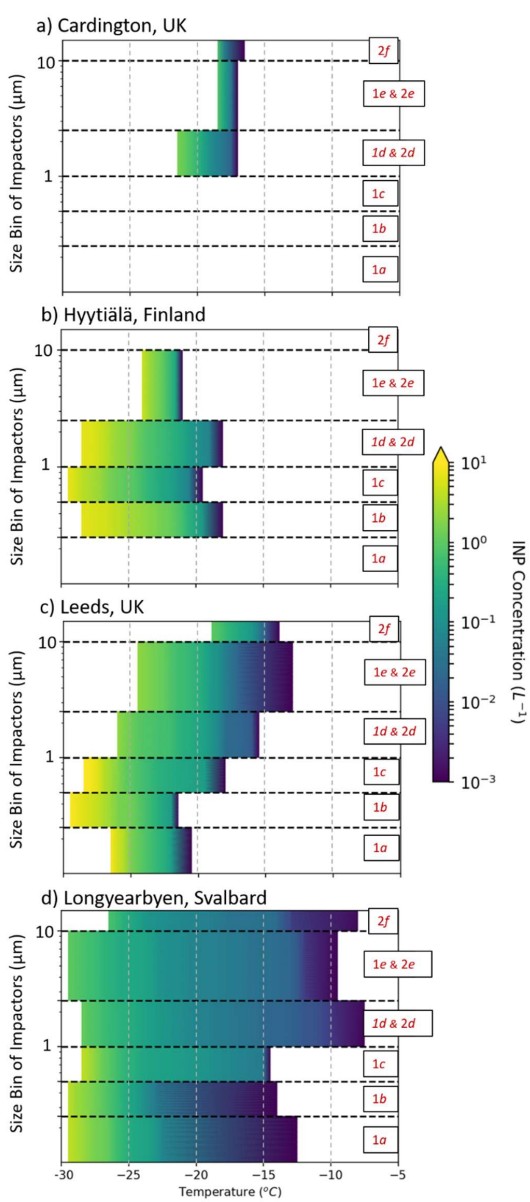

**Figure 9.** Size-resolved ice-nucleating particle concentrations (sr[INP]) for the four test sites: (a) Cardington (UK), (b) Hyytiälä (Finland), (c) Leeds (UK) and (d) Longyearbyen (Svalbard). The colour bars indicate the INP concentration. The dotted lines on the y-axis indicate the size cuts of the impactors.



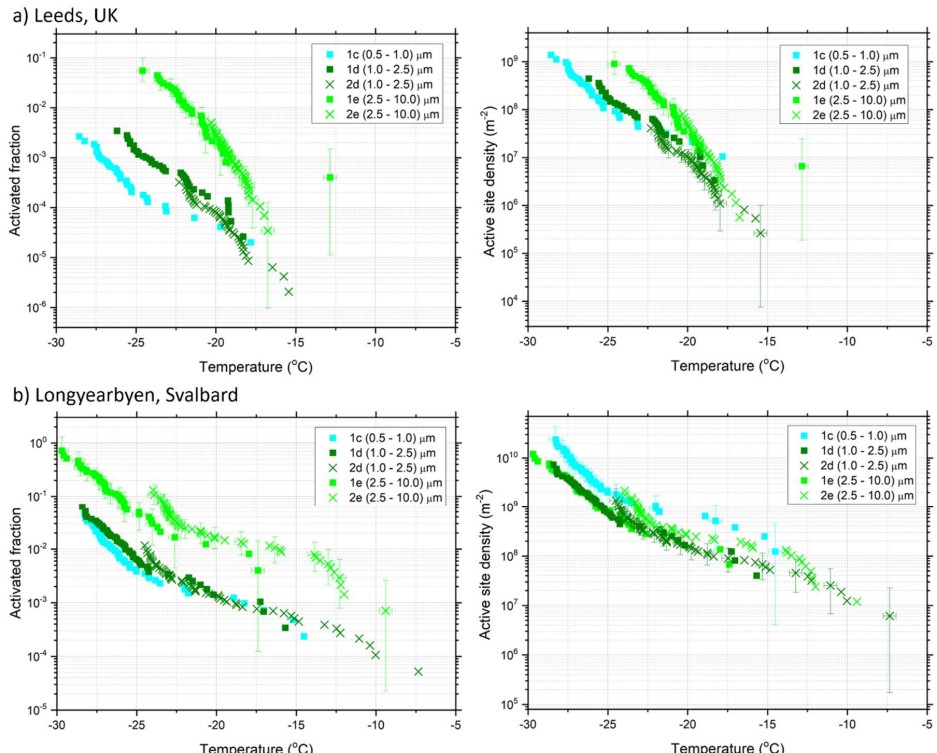

**Figure 10.** Plots showing (left) the activated fraction of aerosol ($n_n$) and (right) the number of active sites per surface area ($n_s$) for samples tested from two measurement sites: (a) Leeds (UK) and (b) Longyearbyen (Svalbard). The colours of the data points indicate the size bins of each impactor, and the different symbols represent the two impactors. Samples of the error bars are shown.