# Peer review of "Resolving the size of ice-nucleating particles with a balloon"

_Atmospheric Measurement Techniques, 2019_

## Referee Comment (RC1) · Anonymous Referee #1 · 15 Jan 2020

This work reports on a new technique for measuring size-resolved INP number concentrations using an aerosol sampling system, SHARK. The SHARK system uses two different types of impactors for collecting aerosol samples for INP measurements. Then, they compare the sized-resolved INP number concentrations obtained using these two impactors and demonstrate that the results were almost the same. This result is very important and valuable, because one of the impactors use a high flow rate up to 100 L min-1, indicating the possibility that this impactor may be useful for collecting sufficient amounts of aerosol samples for INP measurements efficiently during a limited sampling time of tethered balloon measurements. However, it is still unclear whether the SHARK system is indeed a reliable technique for measuring size-resolved INP number

concentrations at high latitudes, because the results presented here are based on only several measurements near the ground level at Hyytiala, Leeds, Longyearbyen, and Cardington. I would strongly suggest that the authors include the results of balloon-borne measurements at higher latitudes (∼100 m above ground level or more) in the revised paper.

General comments:

1) I couldn't find any detailed descriptions regarding a campaign in the High Arctic. When and where were the samples shown in Figures 4 and 5 collected? Why didn't you report any results of INP number concentrations using the SHARK during a campaign to the High Arctic?

2) Because the objective of this paper must be to report a new technique that can measure size-resolved INP number concentrations using a balloon deployable aerosol sampler, the authors need to evaluate the performance of the impactors used for the SHARK system at high altitudes and provide evidence that it can indeed be useful for INP measurements even at higher altitudes (at least, more than 100 m above ground level).

Specific/technical comments:

3) What do you mean by the description "short battery lives of 10s of minutes (Line 79)"?

4) Line 264: "While the particle number concentration increases roughly linearly with size": It seems that the concentrations decrease with increasing the size.

5) Although the authors describe that "the spectra in the four locations have very different characteristics (Lines 345-346)", the spectra may also show some seasonal variations. For examples, recent field studies (e.g., Santl-Temkiv et al., Environ. Sci. Technol. 2019; Tobo et al., Nat. Geosci. 2019; Wex et al., Atmos. Chem. Phys. 2019) show the seasonal variation of INP number concentrations in the Arctic.

6) Lines 358-361: Why did you speculate that the course mode INPs at this site were possibly pollen? In general, pollen grains have much higher sizes (>10 um). Also, why did you rule out the possibility of other possible sources, such as fungal spores, fertile soils, etc.?

7) I couldn't find the results of ice-active site density per mass (n_m) (Line 185), while those per surface area are shown in Figure 10.

8) The authors should provide the more detailed information on the field samplings (locations, time, periods, etc.) at at Hyytiala, Leeds, Longyearbyen, and Cardington in Section 2 and/or table, and not Section 3.

9) Figure 2 caption: "100 $\mu$L min-1" => "100 L min-1" (?)

10) Figure 5: Where were the data from Hegg et al. (1996) obtained?

11) The y axis of Figures 6, 7b, and 8: [INPs] => INP concentration (?)

12) I would like to suggest that the authors show the comparison of INP number concentrations from stages 1d and 2d in Figure 7.

13) Figure 8: I would like to suggest including the information on the size ranges of "Sum of 2d & 2e" and "Sum of 1b, 1c, and 1d" in the figure legend.

14) As also mentioned by the authors, it seems that Figures 8 and 9 are essentially the same. I would like to suggest merging Figure 9 into Figure 8, or simply removing Figure 9 from this manuscript.

15) The y axis of Figure 10: "Activated fraction" => "n_n", and "Active site density" => "n_s" (?)
* * *

---

## Referee Comment (RC2) · Anonymous Referee #2 · 28 Jan 2020

In this paper, the authors describe the SHARK platform to measure the aerosol size and to sample the aerosols for INP analysis. The platform also deploys meteorological sensors. Size-resolved aerosol and INP measurements within the boundary layer are missing, and I think such a platform in the future can be very useful. The paper is well written. I have a few minor comments that I suggest the authors address before the paper can be published.

The importance of aerosol composition towards INP efficiency should be mentioned. Although size is important for transport/dispersion and residence time within the atmosphere; it should be noted that INP efficiency in addition to the size also depends upon

the other factors (e.g., composition: e.g., organics vs. dust, particle type: e.g., spherical vs. non-spherical, etc.). Currently, it reads like size is the most important factor that determines the INP efficiency.

It should be acknowledged that the SHARK technique does not provide spatial and temporal measurements of INP.

It is not clear regarding the use of equation 3 to calculate Ns. Fice and N are determined using different techniques. It is not clear how the measurements from both techniques can be combined. N quantity (line 414) is the total number of particles, which depends upon the volume of air sampled, duration time, and some particle concentration (#/cc). Is it possible that the number of particles that enter the impactor (section 3.2) might be different than OPC (line 415) because of losses within the impactor?
* * *

---

## Referee Comment (RC3) · Anonymous Referee #1 · 28 Jan 2020

The following word in the comment of the Anonymous Referee #1 should be revised as follow.

"high latitudes" => "high altitudes"
* * *

---

## Author Comment (AC3) · 25 Feb 2020

We have adjusted the text of our response to the original comment to reflect this change.

---

## Author Response (AR1)

[revised manuscript text omitted]

**Commented [GP[3]:** Figure changed to show $n_s$ $(T)$ and $n_n$ $(T)$ on y axis.

[Figure]

**Figure 10.** Plots showing (left) the activated fraction of aerosol ($n_n(T)$) and (right) the number of active sites per surface area ($n_s(T)$) for samples tested from two measurement sites: (a) Leeds (UK) and (b) Longyearbyen (Svalbard). The colours of the data points indicate the size bins of each impactor, and the different symbols represent the two impactors. Samples of the error bars are shown.

**Supplementary information for:**

**Resolving the size of ice-nucleating particles with a balloon deployable aerosol sampler: the SHARK**

Grace C. E. Porter[1,2], Sebastien N. F. Sikora[1], Michael P. Adams[1], Ulrike Proske[1,3], Alexander D. Harrison[1], Mark D. Tarn[1,2], Ian M. Brooks[1] & Benjamin J. Murray[1]

[1]School of Earth and Environment, University of Leeds, Leeds LS2 9JT, UK
[2]School of Physics and Astronomy, University of Leeds, Leeds LS2 9JT, UK
[3]Institute for Atmospheric and Environmental Sciences, Goethe University Frankfurt, Frankfurt am Main, Germany

*Correspondence to*: Grace C. E. Porter (ed11gcep@gmail.com) and Benjamin J. Murray (b.j.murray@leeds.ac.uk)

**Contents**

**Table S1.** Details of sampling dates, times, locations and SHARK component information.

| Site | Date (dd/mm/yy) | Sampling period | Impactors | Volume sampled by Impactor 1 at 9 L min$^{-1}$ (L) | Volume sampled by Impactor 2 at 100 L min$^{-1}$ (L) | Impactor 1 after-filter? | Windsond? | OPC? |
|---|---|---|---|---|---|---|---|---|
| Cardington (UK) | 15/05/2018 | 14:15-16:15 (2 h) | Impactor 1 and 2 installed, only Impactor 2 sampling | - | 12,000 | No | No | Installed but not sampling |
| Hyytiälä (Finland) | 11/03/2018 | 10:45-16:00 (5 h 15 min) | Impactor 1 only | 2,835 | - | No | No | No |
| Leeds (UK) | 07/06/2018 | 12:21-15:21 (3 h) | Both impactors | 1,620 | 18,000 | Yes | Yes | Yes |
| Longyearbyen (Svalbard) | 23/09/18-24/09/18 | 20:00-04:30 (8 h 30 min) | Both impactors | 4,590 | 51,000 | Yes | Yes | Yes |
| High Arctic | 20/08/2018 | 10:40-15:30 (4 h 50 min) | - | - | - | - | Yes | Yes |

**S2 Fraction frozen curves for collected samples**

[Figure]

**Figure S1.** Fraction frozen curves for samples collected in Cardington (UK).

[Figure]

**Figure S2.** Fraction frozen curves for samples collected in Hyytiälä (Finland).

[Figure]

**Figure S3.** Fraction frozen curves for samples collected in Leeds (UK).

[Figure]

**Figure S4.** Fraction frozen curves for samples collected in Longyearbyen (Svalbard).

**S3 Aerosol data from sampling in Leeds (UK)**

[Figure]

**Figure S5.** Particle number size distribution data for samples collected in Leeds (UK).

[Figure]

**Figure S6.** Particle surface area size distribution data for samples collected in Leeds (UK).

[Figure]

**Figure S7.** Particle mass size distribution data for samples collected in Leeds (UK).

**S4  Aerosol data from sampling in Longyearbyen (Svalbard)**

[Figure]

**Figure S8.** Particle number size distribution data for samples collected in Longyearbyen (Svalbard).

[Figure]

**Figure S9.** Particle surface area size distribution data for samples collected in Longyearbyen (Svalbard).

[Figure]

**Figure S10.** Particle mass size distribution data for samples collected in Longyearbyen (Svalbard).

**Response to Referee #1**

**We would like to thank the referee for their useful comments and have responded below. The referee comments are highlighted in red and our responses are in black.**

This work reports on a new technique for measuring size-resolved INP number concentrations using an aerosol sampling system, SHARK. The SHARK system uses two different types of impactors for collecting aerosol samples for INP measurements. Then, they compare the sized-resolved INP number concentrations obtained using these two impactors and demonstrate that the results were almost the same. This result is very important and valuable, because one of the impactors use a high flow rate up to 100 L min-1, indicating the possibility that this impactor may be useful for collecting sufficient amounts of aerosol samples for INP measurements efficiently during a limited sampling time of tethered balloon measurements. However, it is still unclear whether the SHARK system is indeed a reliable technique for measuring size-resolved INP number concentrations at high latitudes, because the results presented here are based on only several measurements near the ground level at Hyytiala, Leeds, Longyearbyen, and Cardington. I would strongly suggest that the authors include the results of balloonborne measurements at higher altitudes (~100 m above ground level or more) in the revised paper.

General comments:

1) I couldn't find any detailed descriptions regarding a campaign in the High Arctic. When and where were the samples shown in Figures 4 and 5 collected? Why didn't you report any results of INP number concentrations using the SHARK during a campaign to the High Arctic?

We have added to Table S1 in the supplementary material to present the sample information for samples collected in Figures 4 and 5 including the sampling location and run time for the OPC.

In this paper we have demonstrated the capability of the SHARK payload in four very different locations, at ground level and suspended from a balloon at 20 m, comparing samples from the SHARK with those from ground-based samplers mounted nearby. Data from the Arctic field campaign are shown purely to demonstrate the effective working of all the SHARK sensors. It is not the purpose of this technical paper to present a scientific analysis of INPs at any of the locations from which measurements were obtained, only to demonstrate that the system produces reliable measurements.

During the campaign to the High Arctic we made what we believe to be the first ever airborne measurements of INPs at the North Pole, as part of the MOCCHA campaign. These results are exciting, and we are preparing a separate publication with a detailed analysis, as described at the start of Section 3.1. It is also unclear what these results would add to the current manuscript. We have already shown that: we can sample when the SHARK is in flight, the INP concentrations are consistent with another filter based sampler, we can communicate with the airborne SHARK when it is above 100 m (Figures 4 and 5), and have described the instrument and the analysis in detail. Since there are no other measurements against which to compare the high-altitude balloon-borne samples from the Arctic, they cannot be used to demonstrate the reliability of the system.

2) Because the objective of this paper must be to report a new technique that can measure size-resolved INP number concentrations using a balloon deployable aerosol sampler, the authors need to evaluate the performance of the impactors used for the SHARK system at high altitudes and provide evidence that it can indeed be useful for INP measurements even at higher altitudes (at least, more than 100 m above ground level).

A height of 20 m was chosen for sampling to demonstrate the utility of the SHARK to make balloon-borne INP measurements, whilst providing comparison with another sampler. We chose to work at 20 m rather than 100 m or more in order to have a comparison with a commercial ground based sampler, as this showed that INP measurements made during flight were comparable to those made on the surface.

We have edited the text in Section 3.5 to emphasise this:

"...The Cardington site is an airfield, with large areas of grassy land near a main road, and the sampling was conducted during spring. In order to demonstrate the utility of the SHARK to make balloon-borne INP measurements whilst providing a comparison with a commercial ground-based sampler, the SHARK was sampling whilst suspended from a tethered balloon, flying roughly 20 m above the ground. The INP spectra…"

Additionally, the usefulness of the system should not change with altitude as long as the pumps maintain the flow rate through the impactors. The pumps in the SHARK were chosen because they allow the volumetric flow rate to be maintained while temperature and pressure change with altitude.

We have added the following statement to the text in Section 2.1:

"..used in reverse as a lightweight pump (~120 g). These pumps maintain the volumetric flow rate through the impactors as temperature and atmospheric pressure change with altitude. The pump for Impactor 1 was calibrated to apply this adjustment to at least 2.3 km (Leland Legacy Sample Pump: Operating Instructions, SKC), although the presence of the after-filter may reduce the battery life at this altitude. The pump for Impactor 2 is supplied by a larger battery and should be able to maintain flow to at least the same altitude as the Impactor 1 pump, and over a longer period of time. The SHARK records the volume of air sampled through Impactor 1 during the flight, and so if the pump battery was depleted, or the pressure drop became too great before Impactor 2 had finished sampling, the Impactor 1 pump would shut down and store the recorded value for later analysis. Further testing of the SHARK would be required to define a maximum altitude limit that each SHARK component could operate at. In order to provide RH, temperature…"

Specific/technical comments:

3) What do you mean by the description "short battery lives of 10s of minutes (Line 79)"?

We have re-worded the sentence to read, "...limited by relatively short battery lives, usually under 1 h, and potential…"

4) Line 264: "While the particle number concentration increases roughly linearly with size": It seems that the concentrations decrease with increasing the size.

This has been corrected.

5) Although the authors describe that "the spectra in the four locations have very different characteristics (Lines 345-346)", the spectra may also show some seasonal variations. For examples, recent field studies (e.g., Santl-Temkiv et al., Environ. Sci. Technol. 2019; Tobo et al., Nat. Geosci. 2019; Wex et al., Atmos. Chem. Phys. 2019) show the seasonal variation of INP number concentrations in the Arctic.

We have included the following statement highlighting that there are multiple reasons for the different characteristics of the INP spectra.

"…but the size-dependence is also very different in the four locations. Due to the sample size, these variations could be attributed to the different aerosol population in each location, the time of year and meteorology, which could affect the INP concentrations and spectra (Kanji et al., 2017; Šantl-Temkiv et al., 2019; Tobo et al., 2019; Wex et al., 2019). We now discuss…"

6) Lines 358-361: Why did you speculate that the course mode INPs at this site were possibly pollen? In general, pollen grains have much higher sizes (>10 um). Also, why did you rule out the possibility of other possible sources, such as fungal spores, fertile soils, etc.?

We have adjusted this sentence to read:

"We speculate that the coarse mode INPs at this site were of biological origin, such as fungal material, pollen or bacteria with a steep INP spectrum (Kanji et al., 2017). The steepness of the curve and the temperature are consistent with ice nucleation by pollen (O'Sullivan et al., 2015; Pummer et al., 2012; Tarn et al., 2018). Although the size of whole pollen grains are often larger than 10 µm, pollen is known to release nanoscale materials that nucleate ice, which might be internally mixed with aerosol in this size bin"

7) I couldn't find the results of ice-active site density per mass ($n_m$) (Line 185), while those per surface area are shown in Figure 10.

We have removed $n_m$ from line 185.

8) The authors should provide the more detailed information on the field samplings (locations, time, periods, etc.) at at Hyytiala, Leeds, Longyearbyen, and Cardington in Section 2 and/or table, and not Section 3.

The table describing the field sampling is in the supplementary material section. We do not believe that it fits in Section 2, which is focused on the design and development of the SHARK, but have edited the text in Section 3 to point the reader towards this information:

"…Longyearbyen (Svalbard). Details of the sampling locations, periods, and instrumentation can be found in Table S1 of the Supplementary Information (SI). In this section…"

9) Figure 2 caption: "100 µL min-1" => "100 L min-1" (?)

This has been corrected.

10) Figure 5: Where were the data from Hegg et al. (1996) obtained?

We have added the location that Hegg et al. acquired their data as "Prudhoe Bay, Alaska".

11) The y axis of Figures 6, 7b, and 8: [INPs] => INP concentration (?)

Our convention is to use square brackets to represent concentration, as is done in chemistry. We have done this for previous publications (Harrison et al., 2019, Wilson et al., 2015, Vergara-Temprado et al., 2017), this is also now stated in the figure caption.

12) I would like to suggest that the authors show the comparison of INP number concentrations from stages 1d and 2d in Figure 7.

We have added this to the figure.

13) Figure 8: I would like to suggest including the information on the size ranges of "Sum of 2d & 2e" and "Sum of 1b, 1c, and 1d" in the figure legend.

We have added this to the figure.

14) As also mentioned by the authors, it seems that Figures 8 and 9 are essentially the same. I would like to suggest merging Figure 9 into Figure 8, or simply removing Figure 9 from this manuscript.

While Figure 8 is included as the standard format of reporting INP concentrations, we would like to introduce the srINP plot in figure 9 as an alternative, more intuitive means of presenting data of this sort, and so we have not changed this. We find that when the INP spectra of the different size bins fall on top of each other in Fig 8, Fig 9 makes the distinction between them a lot clearer. We have added the following text to the figure caption of figure 9:

"The data from Figure 8 is presented here in an alternative format, which has the advantage of more clearly and concisely displaying the features of the INP spectrum in each size bin than the plots in Fig 8."

15) The y axis of Figure 10: "Activated fraction" => "n_n", and "Active site density" => "n_s" (?)

This has been changed in the figure.

**Response to Referee #2**

**We would like to thank the referee for their useful comments and have responded below. The referee comments are highlighted in red and our responses are in black.**

In this paper, the authors describe the SHARK platform to measure the aerosol size and to sample the aerosols for INP analysis. The platform also deploys meteorological sensors. Size-resolved aerosol and INP measurements within the boundary layer are missing, and I think such a platform in the future can be very useful. The paper is well written. I have a few minor comments that I suggest the authors address before the paper can be published.

1) The importance of aerosol composition towards INP efficiency should be mentioned. Although size is important for transport/dispersion and residence time within the atmosphere; it should be noted that INP efficiency in addition to the size also depends upon the other factors (e.g., composition: e.g., organics vs. dust, particle type: e.g., spherical vs. non-spherical, etc.). Currently, it reads like size is the most important factor that determines the INP efficiency.

We fully agree that size is not the only determinant of ice nucleation and do not suggest it is. We have adjusted the text in the introduction to read:

"While composition is recognised to be an important controller of ice nucleation ability (Kanji et al., 2017), it has also been generally thought that the larger an aerosol particle, the more likely it is to serve as an INP (Pruppacher and Klett, 1997). However, the lifetime of coarse mode aerosol particles decreases rapidly with increasing size."

2) It should be acknowledged that the SHARK technique does not provide spatial and temporal measurements of INP.

It is implicitly acknowledged that instrument does not produce high temporal resolution, since it is a filter-based technique. However, samples from the SHARK can be used to produce a time-series of INP concentrations with a resolution on the order of hours in specific locations. Similarly, when the SHARK is deployed on a tethered balloon, there are no means of making measurements beyond the length of the tether, but the altitude of the SHARK can be controlled within that range.

3) It is not clear regarding the use of equation 3 to calculate Ns. Fice and N are determined using different techniques. It is not clear how the measurements from both techniques can be combined. N quantity (line 414) is the total number of particles, which depends upon the volume of air sampled, duration time, and some particle concentration (#/cc). Is it possible that the number of particles that enter the impactor (section 3.2) might be different than OPC (line 415) because of losses within the impactor?

The following statement has been added to the paper:

"where $N$ is the total number of particles sampled by the impactor in each size bin, calculated using the number concentration in each size category as measured by the OPC, and the volume of air sampled by the impactor (see Table S1). The size bins from the OPC which have been included in the calculations were matched to those in the impactors. The bin boundaries for the OPC calculations were within tens of nanometres of the impactor bin boundaries."

In addition, we made considerable effort to quantify losses in the OPC and impactors in order to understand them, as demonstrated in Figure 3. We also plot the collection efficiencies for both impactors in Figure 2 and discuss effects such as bounce in the text.

---

## Author Response (AR2)

Dear Prof. Tang

We decided to go with the option B suggested by Referee #1, where we emphasise that we do not show cascade impactor data for SHARK in the free troposphere. As the referee points out, we wish to keep the Arctic tethered balloon data for publication alongside our ship based INP measurements. The Arctic dataset as a whole is very interesting and we cannot jeopardise our chances of publication in a higher impact journal. We can state that the cascade impactors performed as expected at 400 m. Nevertheless, we agree with the referee that we do not present cascade impactor data for the free troposphere, hence should not imply that this is what we do in the abstract and other places in the paper. This has been corrected.

Specifically, we have made the following minor changes:

1. Removing the words "This is especially so in the free troposphere." from the abstract.
2. The addition of the following information on the final line of the abstract - "Test data is presented from four contrasting locations***, with the SHARK sampling at ground level and at 20 m altitude suspended from a tethered balloon, ***showing very different size resolved INP spectra...."
3. In the second paragraph of the introduction, changed "having a lifetime of weeks in the free troposphere" to "potentially having a lifetime of weeks, …"
4. In the second paragraph of the conclusions we now state "Field testing was conducted in four locations close to ground level, and suspended on a tethered balloon at 20 m to demonstrate the capabilities of the SHARK."
5. We have amended a line in the conclusions to include the need to make future measurements at altitude: "It is the intention to make similar measurements in other locations, and at higher altitudes in the future"
6. In addition, in the final paragraph of the conclusions we replaced "The SHARK is an accessible tool for quantifying size-resolved atmospheric INP concentrations through the vertical profile, both within and above the atmospheric boundary layer" with "The SHARK is an accessible tool for quantifying size-resolved atmospheric INP concentrations from a tethered balloon."

Comment on nano-INP:

[revised manuscript text omitted]